# The Initial Questionnaire Development in Measuring of Coach-Athlete–Parent Interpersonal Relationships: Results of Two Qualitative Investigations

**DOI:** 10.3390/ijerph16132283

**Published:** 2019-06-28

**Authors:** Ausra Lisinskiene, Emily May, Marc Lochbaum

**Affiliations:** 1Education Academy, Vytautas Magnus University, 44248 Kaunas, Lithuania; 2Department of Kinesiology and Sport Management, Texas Tech University, Lubbock, TX 79409-3011, USA

**Keywords:** questionnaire, interpersonal relationships, youth sports, coach, athlete, parent

## Abstract

The interpersonal relationship among the coach, athlete, and parent (C-A-P) in youth sports is a complex and dynamic phenomenon. The evaluation of such interpersonal relationship becomes an important factor in trying to understand the overall youth sports environment. The purpose of this study was to begin the development of a questionnaire to assess the coach-athlete–parent interpersonal relationships in youth sports. To achieve our purpose, two qualitative studies were conducted. In the first qualitative study, 136 research participants completed an online questionnaire asking for statements concerning the C-A-P based on an extensive literature review. In the second phase, a follow up phenomenological study design was conducted. For the phenomenological study, 30 participants (10 coaches, 10 athletes, and 10 youth sports parents) completed in-depth interviews. Based on both qualitative study results, the following three themes emerged: group processes, motivation, and over-involvement. The two qualitative investigations revealed themes and 48 potential questions to be used in developing a C-A-P questionnaire in quantifying the C-A-P results.

## 1. Introduction

Grounded in attachment theory, the athletic triad (coach–athlete–parent, C-A-P) concept provides a unique psychological framework and explains the importance of interpersonal relationships between athletic members [1]. First presented by Bowlby [2], attachment theory represents a framework that has contributed significantly to the understanding of the emotional bonds that are formed in early relationships [2,3,4,5]. Attachment theory states that the psychological and behavioral effects of the early parent–child relationship will affect the development of close relationships with other people in the future [2,3]. 

In relation to the sports environment, the investigation by Carr [6] was an important first step in understanding the manner in which adolescents’ internal working models of attachment relate to their friendship quality in the context of youth sport. Secure and insecure parent–child attachment translates into a sport context. The quality of interpersonal relationships between the parent and the child strongly depends on the quality of interpersonal relationships created in early childhood. For example, Carr [6] highlighted that secure attachments in the adolescent–parent relationship are related to positive sporting friendships with teammates and coaches. Davis and Jowett [4] employed attachment theory as a conceptual framework for understanding the interpersonal dynamics within the coach–athlete relationship. Based on the premise that coaches can represent a stronger and wiser attachment figure [7], Davis and Jowett [4] found that insecure attachment styles (i.e., anxiety and avoidant) were negatively associated with relationship satisfaction as well as indices of sport satisfaction, including athletes’ satisfaction with individual performance, training, and instruction, and personal treatment. They concluded that an insecure attachment style presents athletes with greater chances to experience a dysfunctional coach–athlete relationship [4]. However, the athletic dyad relationship can be different from the triad sports relationship.

The concept of the athletic triad often depends on each individuality, on one’s attachment background [6], values that have been internalized [8], the behavior that has been instilled [9,10,11], and on the expectations that are formed [1]. This triad can be stable and educated if all three participants understand certain goals that appear in the competitive youth sports setting [12]. Other cases, however, show a dramatic interpersonal relationship changes, sometimes inappropriate and destructive [13]. In this sense, the C-A-P interpersonal relationship is characterized as a psychological [1,14,15], educational [1,16], and social [1,17] system that occurs within the sports context. A lack of psychological, educational, and social skills in each of the triad member often leads to a complex athletic triad relationship [12]. An unhealthy relationship among all three members leads to a negative sports experience for athletes, coaches, and parents [18].

The psychological aspect of the athletic triad is based on the emotional bond and the attachment between the members and is recognized as being of the highest importance [1]. The understanding of others’ feelings and emotions can lead to a positive interaction amongst all three members. The trust feeling, open and honest communication, and support help to build healthy social relationships that the athletic triad members can trust and rely upon [1]. The educational aspect of the athletic triad is based on the form and the educational methods of communication, sharing of knowledge and understanding each other’s expectations in a youth sports setting. Learning from each other, sharing knowledge, practical experience, and theoretical experience demonstrate the highest triad educational interaction [19]. Social skills are important in maintaining the C-A-P interpersonal relationship and is a vital aspect of effectively communicating and interacting between the members. The important component of social skill development is the understanding that social rules and relationships are created and communicated, but they can change at any time depending on the specific youth sports context. The ability to communicate in a polite and professional manner, to respect each other’s opinion and values, behavior, and attitude are the main components of developing the social skills in the C-A-P relationship [1].

The psychological, educational, and social skills grounded in the athletic triad stabilizes this triangle. A positive or negative triad always affects the young athlete. For example, scientific studies have shown that the interaction between the three parties would be more effective if parents were more involved in youth sport activities and understood the rules that occur within the youth sport context [20], if athletes were not pushed by parents but motivated to play sports by creating a psychologically safe environment [21], and if coaches were provided continuous learning possibilities [22,23]. 

In analyzing the athletic triad, the youth sports triangle or triad is a conceptualization of the total youth sports experience. In understanding the overall triad relationship, the triad includes the roles, responsibilities, and behaviors required for the sports experience to be a success. For instance, coaches are expected to provide a physically and emotionally safe environment for the athletes, provide developmentally appropriate sports experiences, provide an education-friendly environment, and behave in accordance with the coaches’ code of ethics. Athletes are expected to follow the coach’s directions, exhibit good sportsmanship, attend all practices and games, and provide their best effort in both practice and games. Parents should abide by the parents’ code of behavior (e.g., not shouting or criticizing athletes, coaches, or officials), attend as many of the athlete’s games as possible, and ensure the athlete has timely transportation to and from practices and games (when required). Parents provide financial and psychological support but refrain from emphasizing winning over the process of skill and athlete development. Parents should encourage sports enjoyment [24].

Analyzing these dynamic and complex interpersonal relationships poses a need for a deeper analysis and evaluation of the athletic triad relationship in a youth sports setting. The athletic triad has been studied separately by a number of scientists [1,10,12,24] and has been analyzed from the athlete’s point of view [15], the coach’s point of view [19,25,26], and the parental point of view [10,19,27]. However, few studies have attempted to evaluate all three members of the athletic triad within one single study. Studies are lacking, especially qualitative investigations, where all three constructs are combined into one continuous process to enable the examination of the athletic triangle phenomenon. To the best our knowledge, no studies have developed a reliable and valid C-A-P questionnaire that would enable the evaluation of this interpersonal relationship in the individual or team sports context. The scientific literature shows that the development of measurements related to coach–athlete–parent interpersonal relationships is fragmented. For example, the coach–athlete relationship maintenance questionnaire (CARM-Q) was developed by Rhind and Jowett [28]; the coach-created empowering or disempowering coaching questionnaire (EDMCQ-C) was developed by Appleton, Ntoumanis, Quested, Viladrich and Duda [15]; and Sanders, Morawska, Haslam, Filus, and Fletcher [29] developed a questionnaire that assesses the parent and family adjustment processes (PAFAS). The evaluation of the three elements of the system, namely the coach–athlete–parent interpersonal relationships in youth sport, is still unclear. 

We linked C-A-P into a continuous coach–athlete–parent psychological educational and social process and developed a reliable and valid questionnaire for measuring the C-A-P in a youth sport setting. In this sense, the development of a reliable measurement of the C-A-P questionnaire is important and will contribute to a wider audience of science, theory, and practice. The qualitative study results largely contribute and ensure the effective C-A-P questionnaire development process and contribute to the evaluation of the C-A-P interpersonal relationship in team sports or individual sports contexts. The results enable the determination of the exact psychological interpersonal relationship dynamics in the C-A-P. Based on the C-A-P results, intervention programs could be influenced by sports psychologists and sports educators. The teaching of specific skills would help build a positive, healthy, and stable social environment for the C-A-P. Therefore, the purpose of this study was to qualitatively examine the C-A-P interpersonal relationships in youth sports for the initial development of a C-A-P questionnaire.

## 2. Preparation of the Qualitative Investigations

### 2.1. Scientific Literature Review 

First, a scientific literature review was conducted, including a process of collecting, checking, and (re)analyzing data from the existing literature with a particular coach–athlete–parent interpersonal relationship question in mind. The main goal of the systematic literature review was to determine the specific characteristics describing interpersonal relationships in youth sport. The literature review was based on all information found in scientific journals, books, academic dissertations, and electronic bibliographic databases. Electronic databases such as EBSCOhost, Emerald, and Science Direct provided mostly full-text information. A time restriction was applied to the research; we only considered articles published in the last decade: from 2007 (included) to 2017 (included). The main keywords of the analyzed topic were used to generate the systematic literature review: “attachment, sport”, “parent”, “athlete”, “coach”, “social skills”, “interpersonal relationships”, “parental involvement”, and “parental behavior”. All publications were screened by their title and abstract. Those publications not relevant to this review were excluded after this step. The full texts of the remaining publications were read and analyzed in detail, focusing on target participants, and psychological, educational, and social aspects in relation to interpersonal relationships, sports, and only in sports setting. 

### 2.2. Inclusion Criteria 

The articles were included without any cultural or geographical limitation; articles or reviews related to the keywords of interpersonal relationships in youth sports, attachment in youth sports, the coach–athlete dyads, coach–parent dyads, parent–child dyads, parent–coach dyads, and coach–athlete–parent triad topics; articles or reviews related to the any of sports settings (individual or team sports); and articles related to psychology, education, and social interactions.

### 2.3. Exclusion Criteria

We excluded articles or reviews published in a language other than English; articles or reviews that analyzed the role of coach–athlete–parent without considering the aspect of interpersonal relationships; articles or reviews that analyzed the role of coach–athlete–parent without considering the aspect of attachment; and articles or reviews that analyzed the role of C-A-P not contextualized to the sports psychology, education, and social aspects. For example, sports physiology and biomechanics were not included in the systematic literature review process.

### 2.4. Data Extraction and Quality Assessment 

After completing the systematic literature review, the first and third authors independently analyzed the topics. Article titles and abstracts were reviewed. Potential articles were then read next to determine inclusion criteria. 

### 2.5. Results

The search generated 9632 publications. All publications were screened by title, and 1250 articles were excluded due to being duplications. In total, we read the full abstracts of 249 articles. The full texts of 35 publications were evaluated for this study with the target source: attachment/psychology, interpersonal relationships/education, and social interactions/social skills. The articles were included into the data analysis only by the target topic themes. As a result, eight items were found that mainly emphasize the interpersonal relationships within the C-A-P: trust, communication, support, teamwork, respect, motivation, over-involvement, and demotivation. 

## 3. Materials and Methods: Study 1

This study allowed us to determine the meanings of the eight words that emphasize the C-A-P relationship (namely, trust, communication, support, respect, teamwork, over-involvement, and de-motivation), which were generated in the scientific literature analysis. 

### 3.1. Participants and Procedures

The sample consisted of 136 research participants for the preparation study. The research participants included the American and Lithuanian citizens and were recruited from three youth individual and three team sports clubs in the U.S. and four individual sports clubs in Lithuania. Preliminary meetings were conducted with youth sports club administrators, parents, coaches and athletes which served to introduce the research and the underlying procedures to be held at each club. After the club’s approval was given, the schedule and a timetable of the research was planned. Researchers made a careful evaluation of the study’s ethical acceptability. Standard procedures (institutional review and approval, and informed consent) were assured and the study was approved by the Ethical Committee of the Vytautas Magnus University. The main researcher respected the individual’s freedom to decline to participate in or to withdraw from the research at any time. 

With the help of the individual and team sports clubs’ administrators, the questionnaire was completed by coaches, athletes, and parents. A Google forms link was sent out to each of the sports club’s administrators, as agreed. The administrators then shared the questionnaire link to sports coaches, athletes, and parents. The inclusion criteria for the online qualitative study was that the athletes were adolescents aged 12–18. The coaches had at least of two years of coaching experience, and parents had at least of two years of youth sports parenting experience. A Google forms questionnaire was used and was structured as follows. Participants were presented with an approved human subjects consent form. If they agreed and moved to the survey page, each participant self-identified as either a coach, athlete, or parent, and identified the type of sport (i.e., individual or team) when answering the items. There were 26 individual and 92 team sport athletes, 5 individual and 7 team sport coaches, and 5 individual and 11 team sport parents. 

### 3.2. Interview Protocol Development

The eight items, namely trust, communication, support, teamwork, respect, motivation, over-involvement, and demotivation, which were found based on the scientific literature review, were provided to the research participants and asked to explain in one sentence the meaning of the word from the coaches’, athletes’, youth sports parents’ perspectives. The main purpose of this study was to analyze the eight items in greater depth and to find a meaning of each of the item in relation to the overall interpersonal relationship of C-A-P. In this sense, we aimed to analyze, for example what trust means to athlete in thinking of the overall athletic triad relationship. 

### 3.3. Data Collection

The online qualitative investigation involved all three members of the athletic triad. Research participants were selected based on the following selection criteria: homogeneity, information coverage, and informed consent to participate in the survey. All research participants had to be the exact athletic triad at a time. The online questionnaire involved an ethical approval, and the demographic information in relation sports field either individual or team sports. Afterwards, the eight items, in asking to explain each statement, were presented. Parental consent for adolescents to participate in the study was obtained prior to the planned research. We received 136 participant answers on each of the eight statements. In total, 1088 sentences were analyzed.

### 3.4. Data Analysis

We used Google forms platform to gather the information from the research participants. As mentioned above, 136 participants filled the Google forms questionnaire online, thus the transcription was on the platform ready to analyze. In total, 1088 sentences were received with the explanation of each of the eight statements. We analyzed 1088 sentences in compliance with the methodological requirements of content analysis Finfgeld-Connett [30]. First, each of the 1088 sentences were read and re-read by each of the researchers separately. Next, two researchers independently analyzed and coded the first ten transcripts and met to compare their coding and agreed upon a coding scheme. The transcripts were coded to identify meaningful segments of information (i.e., raw data). The researchers followed the eight statements: trust, communication, support, teamwork, respect, motivation, over-involvement, and demotivation. In analyzing raw data, each of the eight statements was analyzed in depth and connected to the sentences that explain a meaning of each fragment. If the sentence meaning were duplicated, the researchers merged and chose the clearest explanation of each statement. For example, when we asked research participants: “In thinking of the CAP relationship, TRUST is when everyone…”, if the research participants’ answers were “can rely on each other” or “can mutually rely on each other actions, words, and beliefs to be for your benefit”, we agreed that these two sentences contained the same information, i.e. “Trust is when everyone can rely on each other”. Following this analysis schema, each researcher analyzed all sentences separately. After each of the eight statement analyses, the researchers met and agreed on a final version of the fragment. In this sense, the researchers met eight times for each of the statement analyses. They agreed on a final version of each statement. As a final result, each of the eight statements involved six explanations (Table 1). 

### 3.5. Ethical Considerations

Research participants participated voluntarily and for no remuneration. They did not receive any misleading information regarding research goals or the form of result presentation. The research was conducted in accordance with the following principles [31]: right to protection from damage, right to safety, usefulness of the study, privacy, confidentiality, and fairness. The ethical principles included obtaining individuals’ consent. The right to refuse to participate in the research at any time was explained. In this sense, the research participant would be excluded from the research, and the answers automatically deleted. 

### 3.6. Quality Assessment

All researchers analyzed the Study 1 data. The minor disagreements were overcome through establishing the sentence order and their fit to the categories. The process was discussed until 100% consensus was achieved. After working independently, we came together and worked diligently until we reached agreement.

Study 2 involved 10 coaches, 10 athletes, and 10 youth sports parents, involving some of the research participants from Study 1 to ensure research quality, which is recognized as the most vital in qualitative research [32]. Thus, to ensure the quality of the structure of the eight categories, we employed the in-depth interpretative phenomenological analysis, which is presented in Study 2.

### 3.7. Results of Qualitative Study 1

The meanings of the eight words were clarified by asking the 136 research participants (coaches, athletes, and parents) to describe each statement in one sentence. The participants described each word. For example, the research participants were asked: “In thinking of the C-A-P relationship, trust is where…”. The research participants described the meaning of the given eight words on the basis of their sports, coaching, or parenting in youth sports experience. The participants generated 1088 sentences for each of the eight categories. From those 1088 sentences, 48 were chosen with six items per conceptualized topic area (Table 1). We then analyzed the sentences in greater depth and removed the duplication items. As a final result, the eight higher-order categories involved six items each in the formation of questions (see Table 1). 

## 4. Materials and Methods: Study 2

For a deeper analysis of the research participants and a follow-up, we used interpretative phenomenological analysis (IPA) [33] in this study to clarify the 48 items found in Study 1 and to merge them into major themes. IPA focuses on the lived experience of participants by incorporating phenomenology and interpretation. IPA is used to analyze in-depth investigations and experiences, such as, in this case, coach, athlete, and parent statements. In total, 30 research subjects participated in the phenomenological analysis. The participants also participated in Study 1. Only those who were interested in the study were involved and invited to be interviewed.

### 4.1. Participants and Procedures

In the second qualitative study, 30 research participants were involved—10 coaches, 10 athletes, and 10 youth sports parents—who responded to the advertisement for volunteers from Study 1. The demographic data of the IPA research participants are presented in Table 2. Demographic characteristics included: the number of research participants; participant education, where all coaches had a higher education degree specifically in sports science, one coach with a Ph.D. degree, six with a master degree, and three with a bachelor. All parents had a higher education degree in different educational fields. All athletes at the time of the interview were studying at high school. In addition, demographics included participant sex, age, sports type (individual or team sports) and sports experience (athletes), sports coaching experience (coaches), or youth sports parenting experience (parents).

The participants in the second qualitative study were recruited from the 136 participants that participated in the first qualitative study, and involved a coach, athlete, and a parent as an actual athlete triad in one interview set. The athlete and the parent were from the same family, and the athlete was coached by the interviewed coach. Based on the eight higher-order statements (Table 1), we conducted an in-depth qualitative IPA analysis to clarify the questionnaire in a final stage that could be used to quantify the questionnaire in the future.

### 4.2. Interview Protocol Development

The questions provided to the research participants focused on the issues raised by the investigation. We followed the 48-item questionnaire presented in Table 1. The main purpose of this study was to analyze the 48 items in greater depth and to generate the item pool in concrete major themes, which, later on, could be quantified with a bigger sample and could finalize the C-A-P questionnaire development process. The interview protocol served only as a guide in each interview to prevent the researcher from uncontrolled deviation from the analyzed topic and to restrict free associations of the participants and the content of the narrative. Semi-structured interview questions began after the lead researcher established consent. Next, the participants shared more about themselves, their families, and the early relationship background with their parents (attachment) and hobbies. Later, more sensitive questions related to the research emerged. The interviews concluded with a neutralizing inquiry about their feelings after the meeting, and an opportunity to ask questions to the researcher.

### 4.3. Data Collection

Lead researcher voice recorded 30 stories of coaches, athletes, and parents. Research participants were selected based on the following selection criteria: homogeneity, information coverage, and informed consent to participate in the survey. With the help of the sports clubs’ administrators, information about the planned survey was announced in 10 sports clubs in Kaunas, Lithuania (seven sports clubs) and New York, USA (three sports clubs). Only those interested in the survey (coaches, athletes, and parents) collected flyers containing the description of the survey and the first researcher’s contact details. Given the sensitive topic and participants’ ages, researcher carefully planned the research process in the following stages. First, researcher obtained ethical approval from Vytautas Magnus University about the eligibility to conduct the research. Next, researcher organized a meeting with each parent and coach of an athlete and provided a detailed explanation of the ongoing research (the ability to accept the invitation, the ability to refuse the participation at any time of the research, and the importance of accompanying the child to the researcher). The parents and adolescents were given all researcher’s contacts, enabling them to ask questions at any time. Finally, parental consent for adolescents to participate in the study was obtained prior to the planned research.

Twelve adolescent athletes, 12 parents, and 15 coaches expressed a desire to participate. However, only 10 coaches, 10 athletes, and 10 parents were included in the study due to information coverage. All 30 respondents were invited for an interview. Interviews were conducted in the first author’s research office in July 2018 in the U.S during her post-doctoral fellowship in New York. The research participants from Lithuania participated in the study in Lithuania in August 2018. The schedule was organized in separate individual meetings with coaches, athletes, and parents scheduled in advance at a time convenient for all parties. Interviews lasted from 50–60 min.

A questions plan was followed to prevent divergence from the research issue and to simultaneously not restrict free associations of the respondents and the content of their stories. Semi-structured in-depth interview questions were asked after informed consent had been established and acknowledged by the researcher to allow the participants to feel comfortable. Next, the participants were asked to share more about themselves, their families, and hobbies. Later, more sensitive questions were asked related to the research subject, for example, “What does trust in the C-A-P relationship means to you?”; “How would you describe a positive C-A-P interaction?”; and “In thinking of the C-A-P relationship, trust is where…”. The interviews concluded with a neutralizing inquiry about their feelings after the meeting and an opportunity to ask questions to the researcher.

### 4.4. Data Analysis

Data were analyzed in compliance with the methodological requirements of interpretative phenomenological analysis [33]. The analysis contained the following stages: transcription, analysis, and credibility checks. Each interview was audiotaped and transcribed verbatim [32]. At this stage, we focused on how the participants talked about themselves: their tone, rhythm, pauses, or changes in topics. IPA requires detailed and comprehensive interview transcription material (text), which was the object of the analysis. Therefore, some essential aspects of participant interactions were noted (laughing, crying, silence, change in mood, etc.). The material was collected in 30 interviews; voice records (more than 30 hours) were transcribed into text. While analyzing the results, the coding system of the qualitative study included changed research participant names, cities, and the exact sports field. The name of the research participant was changed and identified, for example, by “A1” (where A, Athlete; C, Coach; P, Parent; and 1, the number of the interview). The sports field was identified only for individual or team sports without any explanation of the type of individual or team sport. The researcher followed qualitative research ethical requirements [32,33] to ensure and maximize the security of the research participants’ identities. 

The steps in the qualitative analysis began with reading, re-reading, and writing memos. While reading, we wrote memos by taking note of initial observations and developing phenomenological comments. The next step involved coding the data by segmenting and labeling the text development of arising topics. Using codes, we developed themes by aggregating similar codes and then looked for the connection between the main topics to interrelate themes. At this stage, another participant joined the analysis, and the process was repeated from the beginning for each case. The cases were treated individually prior to looking for interrelations between cases. At this stage, the thematic structure of each individual phenomenon was briefly described, converting it into a written interpretation of the phenomenon. 

All the analyzed topics were interconnected and analyzed within psychological, educational, and social aspects. The psychological aspect (the attachment) involved detail and careful attention in analyzing the results as the C-A-P feeling of trust, support, and motivation formation. The over-involvement and demotivation constructs were also included under the psychological framework. The educational aspect included careful attention to the respect construct, which can be taught. The educational aspect also involved the team-building construct where the members are constantly interacting and using educational methods while interrelating. The social aspect included careful attention when analyzing the communication construct. The schematic structural form was transformed into a narrative, highlighting the difference between a participant’s narrative and the researchers’ interpretations [33,34]. In the final stage of analysis, the credibility of the findings was ascertained by triangulating different sources of information, member (participants) checking, inter-coder agreement, rich and thick descriptions of the cases, and reviewing and resolving disconfirming evidence [35].

All 30 coach, athlete, and parent interviews were analyzed separately, and only then were merged into one single study to determine the commonalities in the C-A-P relationship. 

### 4.5. Ethical Considerations 

Research participants participated voluntarily and for no remuneration. They did not receive any misleading information regarding research goals or the form of result presentation. The research was conducted in accordance with the following principles [31]: right to protection from damage, right to safety, usefulness of the study, privacy, confidentiality, and fairness. The ethical principles included obtaining individuals’ consent. The consent included information about the ongoing research. The consent included information about the research participants’ changed names, city, and exact sports field. The right to refuse to participate in the research at any time was explained. In this sense, the research participant would be excluded from the research, and the recorded interview automatically deleted. The researcher followed the four principles of research quality assessment in [36]: (1) sensitivity to context; (2) commitment and rigor; (3) transparency and coherence; and (4) impact and importance. 

### 4.6. Results of Qualitative Study 2

The results of the designed C-A-P concept are presented in Table 3. Three major themes emerged in relation to coach, athlete, and parent interpersonal relationships in youth sports: group processes, motivation, and over-involvement. These three major themes characterize the C-A-P interpersonal relationships and are the key components in evaluating the C-A-P relationship in a youth sports context. This qualitative study was one part of the C-A-P questionnaire development process. The three key themes that emerged will be then presented in quantitative data, which would enable the development of a reliable and valid C-A-P questionnaire for measuring the coach, athlete, and parent interpersonal relationships. 

*Group processes.* Based on research participants’ stories, the group processes were merged into psychological, educational and social aspects as they are closely related to each other. The conversations with coaches, athletes, and parents revealed that the common goal could be reached within the C-A-P through open and honest communication, support, teamwork, trust, and respect. All five components were united and described the C-A-P major theme: group processes. The C-A-P concept is seen through group processes and is described as a team where all the participants reach the same goal. However, the way in which group processes work could be explained through the subthemes, which explains the overall group processes concept.

*Communication.* All three members highlighted communication as the key component in enhancing the relationship in the C-A-P as a group process. Communication enables individuals to socialize, interact, communicate, and interrelate. Social skill development through the communication is one of the main components in building the relationships. However, as the research participants noted, communication can be more or less effective. The most important aspect in C-A-P communication is to be heard and understood, even if every person has their own idea or opinion. Positive communication occurs when all three members talk openly and honestly. The research participants noted that the C-A-P need to clearly understand each other, express their own ideas, thoughts, and opinions, and must be comfortable with the conversation. In contrast, negative or poor communication occurs when at least one member of the athletic triangle is too criticizing or too demanding and complains in an inappropriate manner. Healthy communication is not where everybody should agree with one another’s opinion or suggestion; conversely, healthy communication occurs where all three members are able to express their ideas and a final decision or agreement can be made without anger and criticism: “A problem occurs when one of the C-A-P members do not express their opinion openly. Fear, the feeling of the doubt...To express the ideas openly and willing that you will be heard is important in the C-A-P relationship” (P-5); “The key component in C-A-P is communication, and it is open but therefore, the participants must know the hierarchy of communication. Players to coaches, players to parents. Then coaches to parents and vice versa” (C-5); “Clear communication via email/ post-game comments, and during coaching sessions enables solving all problems” (P-1); “When everyone can openly speak out about concerns or anything in general and feel comfortable” (A-3); and “The communication is positive only when everyone is comfortable discussing issues, success, or changes in a civilized manner” (C-2). Effective communication, as one of the research participants mentioned, could be attained through enhancing the dual role of the C-A-P. By reflecting and interpreting the research participant coaches’ thoughts (see quote C-5), first, the members should clearly understand the dual communication: the coach–parent, coach–athlete, coach–athlete, and the athlete–parent. When the dual role is accepted, then the C-A-P is possible. It was interesting to hear the idea that only an effective dual communication relationship enables building stable triad communication. 

By interpreting the results, we found that a successful relationship between all three participants is vital. However, the meaning of the phrase “successful” from all three C-A-P members’ perspectives could be perceived differently. Scientific studies showed that a successful and effective athletic triad is a challenging task in youth sports because all three members of the C-A-P have different values, behavior, and attitudes. The relationship between two members is not the same as among the three members. To find the best method of communication and positive interaction is a difficult task for each of the three participants. 

*Support.* There are numerous psychological aspects in relation to youth sports and C-A-P interpersonal relationships. As an emotional feeling, support is important to surviving in a competitive sports context, especially to athletes. The warmth from parents and the supportive atmosphere surrounding the coach make the athlete feel safe and stable. However, the C-A-P is a three-way relationship, and each person understands support differently. Every component of a relationship directly reaches and directs every person. The ability to identify one’s feelings is the key component in the C-A-P relationship. Support of coaches from parents is necessary in a different form. For example, after successful or unsuccessful sports competitions, the parent and the athlete still trust and support the coach, and understand that, in the sports context, victories and defeats are necessary for the overall athlete development: “Support is when coaches know that parents believe in them and not trying to coach their children. For an athlete, feeling support is simple little things—parents buying new shoes, asking is everything going okay, and most important, when parents let them believe in their dreams (most athletes feel pressure from parents because they say it is not possible for them to achieve professional athlete level)” (A-6). “Encourages and supports the communication and trust as well as creating an encouraging environment for everyone to excel” (P-7). “Cares for one another and backs up each other” (P-2). “Feels like they have others around them that are building them up and holding them up at all times” (C-4). A supported atmosphere is important aspect in the C-A-P group processes and should be enhanced by all three C-A-P members.

*Teamwork.* Developing a strong overall team culture by creating unity and strength amongst the three members can lead to a powerful C-A-P. The team culture explains the overall rules and goals within the C-A-P interpersonal relationships. As the research participants noted, building the team largely depends on the coach. The coach is one of the main figures that can model such culture. The only question arises is in the ways through which the coach develops such culture. Based on research participant stories every person should feel an equal part of the C-A-P, by their decisions, feelings, suggestions, and ideas, and should be working toward the same goal. Based on the research participants’ stories, teamwork could be characterized as when everybody works hard together to achieve a goal, during the game everyone is moving and thinking as one for the team, communicates, plays well together, and everyone gets along: “Works together toward achieving a common goal as a team, not an individual. At the same time, everyone knows their role” (C-3); “A strong relationship appears when C-A-P is attached to each other. Trust, support, help at all times. Feels safe and secure within this relationship” (P-1); and “Work together to accomplish something bigger than any one of them can achieve alone” (A-9). Social networking, social skill building, and working together in reaching the same goal describe the construct of teamwork. Therefore, by interpreting the results, the psychological sound could be felt in relation to attachment. In long-term C-A-P, the members become attached and feel emotional bonds to each other. As one athlete noted, “I was about to quit the sport, I realized that it was not fear because of the close relationship and the bond between us”. In this sense, attachment theory helps to explain such phenomena. The secure attachment to the caregivers (parents) or significant others (coaches) that was built leads to a very strong emotional bond that lasts a life time. 

*Trust.* Being able to trust an adult outside of one’s family can be a powerful force in an adolescent’s life. Unfortunately, there is frequently a lack of connection between adolescents and adults in our society. Positive relationships with adult leaders, such as coaches, in youth programs appear to positively impact youths’ physical, emotional, social, and moral development [37,38]. Often, coaches, athletes, or parents do not recognize the power of their words or of the implied messages they send. In building trust, every action or word can be received differently, which will affect the person either in positively or negatively. At this moment, trust is formed. The feeling of trust ensures safety in every individual. The feeling of trust can maximize open and honest communication, support, and connection, which, in long-term relationships, appear as an emotional bond, which is the so-called attachment. Trust is a powerful feeling that leads to positive communication and effort to be better: “The feeling that you can always rely on, seems as very important factor. Only then you can reach more than you are able to” (A-1); “You can depend on each other for help whenever needed. You feel safe, you express your thoughts, and can ask for something at any time” (P-8); “Can rely on one another to be there for them no matter what differences may arise” (A-2); “Can mutually rely on each other actions, words, and beliefs to be for your benefit” (A-9); and “Trust, one of the most important components of the C-A-P that unites the three members” (C-5). 

Adolescent athletes may place trust in a coach or parent based on whether the person is reliable in keeping a promise, whether the person refrains from causing emotional harm, such as being receptive to disclosures, maintaining confidentiality, refraining from criticism, and avoiding acts that elicit embarrassment. In remembering the reflection of C-A-P conversations from the athletes’ point of view, it is very important to build the feeling of trust, and it is the most important component in building a positive and stable C-A-P relationship. If the feeling of trust is not present among the C-A-P, all efforts to rebuild trust are ineffective. As one professional athlete pointed: “No trust, no relationship, there is no longer C-A-P, and the game is over” (A-1). For example, if the communication could be enhanced, teamwork could be built, respect could be taught, and based on our participants’ stories, the trust feeling could hardly be rebuilt. 

*Respect.* The sports environment is a great place to grow and establish respect. While involved in sports, participants learn the importance of respecting their teammates, coaches, parents, opponents, and spectators. If an athlete wishes to improve and succeed, they must listen to the advice and criticism of those who coach them. Part of being respectful is understanding others may know better and we should listen and learn from them. Respect appears when every C-A-P member treats each other equally, respects every member as the most important figure of the system, and are the authority over one another. If there is no respect, there is no communication and no enthusiasm to reach for a common goal: “When people take other’s opinions, values, and insight, and acknowledge these opinions because everyone is different, everyone brings a different perspective to the relationship and those different perspectives can be used to make the environment the best it can be” (C-3); “Holds the others in high esteem and treats each other like they are equally important” (P-4); “Be respectful, but understanding that it is a sporting event and coaches yell and curse, but it is motivational, not bad” (C-8); “Understands and acknowledges others’ opinions, but also understands that there is no compulsion to think the same and can discuss the conflict in a civil manner” (P-10); and “Is separate, but equal. You trust that each person is doing their respective jobs and at least make sure you are doing yours” (A-2). 

*Motivation.* The research showed that the expression of passionate involvement in the C-A-P, including hard work toward the same goal and simultaneous enjoyment of the chosen activity, emphasizes motivation. Motivation is important in every athlete’s sporting career and in everyday sporting life. Motivation is important in coaching and parenting experience as well. The motivation of each of the C-A-P members enables building a powerful and empowering motivational sports climate. All three participant groups expressed such aspects that describe the overall motivation of the C-A-P: “Wants to do the best they possibly can, and giving 100% effort the whole time” (C-10); “Coach is the middle connector of athletes and parents. If the coach connects them, there will be motivation for both athletes and parents” (P-4); “Has the same goals and ambitions, which then help everyone work hard in the weight room and in practice”; “Everyone knows what they want to get out of a certain sport or position” (A-7); and “Shows up with positive attitudes and works to the best of their abilities” (C-3).

*Hard work.* In the sports arena, working hard to achieve a common goal is essential. A talented person without putting in effort, hard work, and patience cannot succeed. Based on the research participants’ responses, hard work is defined by when everyone remains positive and encourages each other to do/be their best and everyone is working hard to achieve the common goal: “Wants the win” (A-10); “Wants to do the best they possibly can, and giving 100% effort the whole time. The desire to be better than they are” (C-10); “Is enthusiastic about being involved in the sport” (A-2); “Motivation is present when shows up willing to do their part and put their best effort forward, when everyone is working hard toward a common goal, no matter the day they are having. This can be an area where communication is essential” (P-9); and “Has the same goals and ambitions which then will help everyone work hard in the training sessions and in competitions” (C-1).

*Passion.* Coaches, the athletes, or the parent contribute their love to the chosen sports activity. The coach is a professional that contributes their passion to their chosen job. The coach is educated and innovative and seeks to provide C-A-P sports education, but not to produce the best sports results. Sports results are important in a sports environment, but understanding the overall priority of sports education, the psychological aspect, and the social aspect is essential. The more important aspect of being a coach or the parent what the athletes have learned from the messages they have heard, that they have received, and the values they have learned about being an athlete. The athlete is passionate when is waiting for training sessions and the excellence they can achieve in training sessions and in competitions. The athlete is happy with the chosen activity and can express themselves through it. The parents are positively involved and show commitment and joy in being the youth sports parent: “Training sessions with excitement, parenting with support and praise, coaching with emotion” (P-7); “Is excited and glowing from head to toe” (P-3); and “Inspires each other to be great at all times” (A-3). 

*Enjoyment.* Enjoyment is one of the most important and positive emotions in youth sports. Sports enjoyment is a positive affective response to a sports activity that reflects feelings such as joy, happiness, pleasure, and fun. The joy of being a part of the sporting community, practicing in training sessions, and competing, the feel of the victory, and overall athletic satisfaction lead to the most positive emotions, and it is an important aspect of continuity of such activity. Demonstrating superiority and success intrinsically influences a young athlete’s motivation, which is a key predictor of sports commitment. Enjoyment within all three members of the C-A-P system could lead to an effective and successful C-A-P relationship: “Natural emotions, such as the joy of being the athlete, the joy of being the youth sports parent, the joy of being the coach” (C-2); “Love of the chosen activity and living through it” (P-5); “Attachment to each other (the C-A-P) and the warmth that could never be replaced” (P-2); and “The joy in being a part of C-A-P and the pride in each other always makes positive” (A-3).

*Overinvolvement.* As the sports environment is a positive social environment where athletes’ values are instilled, the behavior formed, the emotions expressed, and the relationship between the participants involved in sports can vary dramatically. For example, dominance in the sports environment can lead to a number of positive outcomes, such as motivation, achievement of goals, victories, etc. However, in relation to C-A-P interpersonal relationships, excessive dominance means over-involvement, which influences negative feelings toward the C-A-P members. Research participants stated that these are actions far outside acceptable limits. Such relationships in C-A-P describe the situation in which one person holds more power, influence, or success than others. The relationship among the C-A-P then becomes demotivating, controlling, and demanding.

*Demotivating.* As motivation refers to athletic, coaching, or parenting behavior that is driven by internal or personally meaningful rewards, for example, opportunities to explore, learn, and actualize potential, demotivation occurs when at least one member of the C-A-P does not really want to be a part of the C-A-P. They do not have goals or certain standards they want to accomplish, they do not care about what happens and wants to invest the least effort required. The C-A-P are not communicating with each other, are silent, and have no interest: “Does not want to do the work no matter what the outcome” (C-5); “Is not really wanting to be there as a coach, an athlete, or parent. They do not goals or any certain standards they want to accomplish” (P-8); “Is not having fun and lacks the desire to reach the ultimate goal/win” (A-3); and “Treats the athlete, the coach, or parent with disrespect. Stops trusting and believing in the team” (P-2).

*Demanding.* The degree of difficulty, level of risk, and fitness requirements (strength, power, coordination, endurance, agility, flexibility, etc.), and the time commitment are challenging issues and affect all three members of the C-A-P. Naturally, sports can be demanding in different ways. Being surrounded by C-A-P interpersonal relationships is another demanding task. Based on research participants, demand in sports is different from demand in C-A-P relationships. The demanding relationship is emphasized as a negative feeling based on participants’ stories that often occurs within the C-A-P relationships as overstepping boundaries, being controlling, and being overwhelming: “Intervening at unnecessary times or taking on positions that someone else already has, such as a parent trying to coach the players along with a coach” (C-6); “Trying to do another’s job. A coach cannot be a player. A player cannot be a parent, and a parent cannot be a coach” (A-1); “Overstepping the boundaries or responsibilities of another party involved” (P-2); “Being critical without being constructive” (C-10); “Crossing boundaries for their own agenda that they believe is right” (A-9); “When someone tries to tell someone else what to do in a rude manner instead of as a teammate” (A-1); and “Focusing on one aspect too much (either coach is focusing too much on one person or parent is focusing too much on coach etc.)” (P-5).

*Controlling.* Controlling behavior can be a part of C-A-P. Sometimes the purpose of one person’s behavior is to control and intimidate others. Such people are critical, controlling, manipulative, and use abusive or destructive behavior against others and seek a specific goal. The coach could be overinvolved and controlling with strict requirements and focusing only on winning and best performance. Parents could also influence their child because of their goal for their child to be the best in the sports arena without knowing that they are using a destructive and inappropriate method of communication. Either way, the coach or the parent will dominate the process or often the unhealthy relationship among the C-A-P occurs, which in turn will lead the athlete to drop out or burn out: “Hindering your success or growth to a point that you are unable to continue with that influence” (A-1); “Not on the same page. One person is overstepping their bounds, such as when a parent tries to take on the role of the coach from the side lines and starts to overrule the coach or disagree” (C-5); “Gets angry, loud, and/or disrespectful. Constantly shouting of the side lines over the coaches directions” (A-4); “Way too involved in the relationship, making it difficult for the others to perform their duties and causing strain on the relationship” (C-2); “Too passionate and does not know how to handle that emotion” (C-1); “Overly emotionally invested in the sport and finds a hard time separating life from the sport” (C-3); “Dominating the process. Trying to do someone else’s job” (A-8); and “Not working with the team, trying to do their own thing and will not listen to other people or their ideas on how to help the team succeed” (C-1).

## 5. Discussion

The purpose of this study was to examine coach-athlete–parent (C-A-P) interpersonal relationships in youth sports to develop a C-A-P questionnaire. Two qualitative studies were employed with a goal to finalize the main dimensions that could be used in the development of a C-A-P questionnaire. 

Prior to the qualitative studies, a systematic literature review was conducted to identify what specific words emphasize the overall C-A-P interpersonal relationships. Based on the literature review we found eight main statements that emphasize the overall C-A-P relationship: trust, communication, support, motivation, respect, teamwork, demotivation, and overinvolvement.

The first qualitative study was conducted to finalize the eight higher-order categories. The eight words were sent out again to the same 136 participants (coaches, athletes, and parents) to determine what each of the words means to them, from the coach, athlete, or youth sports parent perspectives. The participants generated 1088 sentences for each of the eight categories. From those 1088 sentences, 48 were chosen with six items per conceptualized topic area (Table 1). The second interpretative phenomenological analysis was completed to analyze the study results generated from Study 1. Three themes emerged from the IPA study: group processes, motivation, and overinvolvement (Table 3).

First, the major theme “group processes” emerged, which covers themes such as communication, support, teamwork, trust, and respect. Conversations with research participants revealed that the C-A-P triad works together as a team and involves all C-A-P participants when solving specific issues such as defeats, trauma, competition fear, and failure together. The joy of victory and the achieved successes, the plans and setting goals for the future must be communicated together, with all C-A-P members involved, not separately. 

Our main task in developing the C-A-P questionnaire was to find the main constructs that emphasize the overall C-A-P relationships. Based on the research participants’ stories, we merged the psychological, educational, and social constructs into one single phenomenon: communication, support, teamwork, trust, and respect, and named the main major theme “group processes”. In trying to understand the overall group processes phenomenon, the psychological framework of attachment theory helped us to explain that the interactions between the members are important for maintaining the relationship between the members. To maintain a close relationship as a group or a team, communication and respect must be present and the feeling of trust is essential. A supportive atmosphere surrounding the young athlete, created by the coach and parents, creates a safe and secure attachment amongst the members. A group, as a social environment, interacts and interrelates, and uses different methods for communicating. Over the long term, such group members become attached to each other. In line with this, the findings reported by Van Puyenbroeck Stouten and Vande Broek [39] suggest that a need-supportive coaching style enables coaches to create a mastery climate. This climate seems to encourage athletes to be proactive, which enables teams to effectively tackle encountered challenges. Fry and Gano-Overway [40] suggested that coaches be provided with strategies for creating a positive and supportive climate in sport.

Strategies are provided to help coaches to foster motivation from the first day they meet their athletes. Specifically, these strategies focus on helping athletes get to know each other better, gauge their success based on their effort and improvement, and bringing parents on board to be part of their children’s experience. Research has highlighted that coaches have wonderful opportunities to educate parents when they work with athletes and to help promote a positive approach to the sports environment. In relation to C-A-P, it is important that the constructs of communication, support, teamwork, trust, and respect are seen as one phenomenon in the formation of positive group processes in the C-A-P. The strategies provided by Fry and Gano-Overway [40], Gano-Overway et al [41] and Smith, Smoll, and Cummings [42], in relation to teambuilding, can help coaches send a strong message to athletes and their families that efforts are being made to create a positive and supportive environment for the team. The research has highlighted how beneficial such an environment can be for athletes. For example, studies have revealed that perceptions of a caring and task-involving climate (i.e., one focused on participants’ individual effort and improvement) are associated with children having more fun, trying harder, demonstrating better sportspersonship, experiencing less anxiety, and reporting a better ability to handle their positive and negative emotions. Additionally, such climates are linked with athletes interacting more positively and engaging in more caring behaviors with their coaches and teammates, and expressing more empathy for their peers [40,41,42].

The second major theme that emerged from our study is motivation, which emphasized the overall involvement of all three members of the C-A-P. All participants stated that motivation is important to achieve the common goal. Motivation is connected to smaller themes, such as hard work, passion, and enjoyment. Enjoyment is important in the continuity of sport; however, without hard work, there would be no motivation to achieve and enjoy the activity, and to achieve the common goal of the C-A-P. In discussing the C-A-P motivation theme, for instance, Appleton, Hall and Hill [43] examined the effects of parent-initiated and coach-created motivational climates on athletes’ perfectionistic cognitions and revealed that the parent-initiated motivational climate is a significant predictor of athletes’ perfectionism-related to thoughts. The results provide support for the influence of the coach-created motivation climate over children’s perfectionistic cognitions. Duda and Appleton [44] observed empowering and disempowering features of the multidimensional motivational coaching environment in training and competition in youth sport, and found that coaches were observed to create a less empowering and more disempowering environment in competition compared to during training. One of their findings recommends future C-A-P research because it could be possible that the interpersonal relationships between the C-A-P members could differ in relation to the pre-competition and in-competition environment, but again, there is a need to test such results after the C-A-P questionnaire development process complete. 

The final theme that describes the overall C-A-P interpersonal relationships is over-involvement. As sport, by its very nature, is a social context where participants socialize, interact with, relate to, and influence one other [11]. Sport participants have the opportunity to explore and develop their cognitive, educational, and social selves [6], and attachment becomes of prime importance in the lives of adolescents who are passing through a period of psychological, educational, and social transition between childhood and adulthood [45]. However, in sports environments and in studying the interpersonal relationships in sports, the interactions are not always positive. In terms of attachment theory, insecure attachments form when people are not physically or emotionally present. In these situations, either insecure-anxious or insecure-avoidant attachment styles form. In the C-A-P relationship, if overinvolvement occurs, demotivation, criticism, and negative emotions toward the young athlete are displayed, and in long-term sports development, the athlete will reject such involvement from either the coach or parent.

Research participants talked about the positive interpersonal relationships that occur within the triad; however, the research participants mentioned that negative feelings and behaviors occur as well. The triad relationship is complex and dynamic, especially during adolescence, requiring psychological, educational, and social skills from the coach and parent in working with young athletes. It also requires more time commitment for specific communication strategies, understanding, and supporting the athlete. Two-way communication is different, as three-way communication requires more patience, understanding, and communication skills. Therefore, the over-involvement theme covers subthemes such as demotivation, demanding, and controlling in the C-A-P context. As sports coaching is a complex social and dynamic endeavor, where coaches have to interact with athletes, other coaches, and parents [46], the interaction is not always positive. For example, Kristiansen et al. [46] examined the careers of two successful female elite athletes who later stagnated and identified possible factors that might have led to their demotivation and found several issues related to coaching and coach education. Their individual profiles revealed that their perception of the lack of long-term development was caused by coach miscommunication, having to cope with sudden fame, and injuries provoked by overtraining. In addition, our study showed that C-A-P overinvolvement, demotivation, and demanding and controlling behavior often lead to athlete dropout or burnout. For example, Montesano, Tafuri and Mazzeo [47] found that the coach does not have to be an authoritarian leader, authoritative, and not too permissive, but instead should be empathetic, motivating, stimulating, and enthusiastic. Our study results showed that the C-A-P interpersonal relationship could be stabilized through effective group processes, enhancing motivation, and avoid overinvolvement in youth sports. However, in developing the C-A-P questionnaire, the main constructs that were revealed (group processes, motivation, and over-involvement) need to be tested by applying the quantitative study with a bigger sample of participants.

## 6. Limitations and Conclusions

This study has two main limitations. First, our study involved participants expressing themselves concerning the overall sports experience in relation to the C-A-P relationship. Research on the interpersonal relationships separating the pre-competition season, in-competition season, and post-competition season is needed. Second, the attachment style that was developed in the early years was unknown. It would be beneficial to conduct in-depth psychological research in trying to determine particularly secure or insecure attachment styles that were formed in the early years of the athlete. Second, our study was not a longitudinal study where C-A-P relationship changes could be observed and evaluated considering a specific time frame, especially regarding the athlete’s early, middle, and late adolescence. Even with these few limitations, taken together, the qualitative results of the C-A-P reveal that the three themes (group processes, motivation, and over-involvement) that emerged could be used in developing a C-A-P questionnaire as the main dimensions in further quantifying the C-A-P questionnaire. In addition, based on this qualitative C-A-P investigation sports practitioners, coaches, sports psychologists could use this study as a background in understanding the overall C-A-P relationship and in some cases start implementing the intervention programs in solving C-A-P negative processes.

## Figures and Tables

**Table 1 ijerph-16-02283-t001:** Coach–athlete–parent (C-A-P) list of the original 48 items.

Dimension	Initial Questions
Trust	1. My CAP relationship is reliable 2. In my CAP relationship, everyone believes in each other 3. In my CAP relationship, everyone is honest with each other 4. In my CAP relationship, you can depend on each other 5. My CAP relationship is based on confidence 6. My CAP relationship provides a safe environment
Communication	7. In my CAP relationship, everyone is heard 8. In my CAP relationship, everyone talks openly 9. In my CAP relationship, everyone talks honestly 10. My CAP relationship allows everyone to express themselves 11. In my CAP relationship, everyone understands each other 12. In my CAP relationship, everyone knows what is expected
Support	13. In my CAP relationship, we all can depend on each other 14. In my CAP relationship, everyone helps each other through wins and losses 15. In my CAP relationship, everyone cares for one another 16. In my CAP relationship, everyone can rely on each other during hardships 17. In my CAP relationship, everyone helps each other 18. In my CAP relationship, we encourage each other
Teamwork	19. My CAP relationship concerns all involved in achieving a common goal 20. My CAP relationship understands that everyone in the CAP is important for success 21. My CAP relationship has the same clear understanding of what the goal is and how we as a team are going to reach that goal 22. My CAP relationship believes in each other as a team 23. In my CAP relationship, we are attached to each other 24. In my CAP relationship, we have a bond with one another
Respect	25. In my CAP relationship, everyone values each other 26. In my CAP relationship, we see one another as equals and vital to the team 27. In my CAP relationship, everyone feels appreciated 28. In my CAP relationship, everyone feels important 29. In my CAP relationship, everyone treats each other with the same respect 30. In my CAP relationship, we are mindful of others opinions
Overinvolvement	31. In my CAP relationship, one member in the relationship expects more 32. In my CAP relationship, one member is overstepping the boundaries of another person’s role 33. My CAP relationship is being controlled by one of the CAP members 34. In my CAP relationship, one member is too demanding 35. In my CAP relationship, one member is too concerned 36. In my CAP relationship, one member is doing more than is required
Motivation	37. In my CAP, everyone feels driven/passionate to achieve the common goal 38. In my CAP relationship, everyone involved is enthusiastic 39. My CAP relationship stays positive and encourages each other to do their best 40. In my CAP relationship, everyone works hard to achieve a goal 41. In my CAP relationship, everyone is willing to do their part and their best 42. In my CAP relationship, everyone is trying to be better than they are
Demotivation	43. My CAP relationship is lacking a common goal 44. In my CAP relationship, no one has any interest in working together 45. My CAP relationship does not show effort 46. Those in my CAP is not equally motivated to work for the improvement of the team 47. My CAP relationship is not cohesive in achieving a common goal 48. Those in my CAP relationship give up on each other and the team falls apart

**Table 2 ijerph-16-02283-t002:** Demographic characteristics of research participants (Study 2)**.**

Participant	Education	Sex	Age	Sport Type	Sports Experience; Sports Coaching Experience; Youth Sports Parenting Experience
10 Coaches	Higher	7 women 3 men	45	5 Individual 5 Team Sports	10–15 years
10 Athletes	High school	6 girls 4 boys	17	5 Individual 5 Team sports	9 years
10 Parents	Higher	5 women 5 men	50	5 Individua l5 Team Sports	9 years

**Table 3 ijerph-16-02283-t003:** The C-A-P phenomenology study theme table.

Major Theme	Theme	Subtheme
**Group processes**	Communication	Honesty
		Expression of ideas
		Understanding
		Three-way communication
		Everyone is heard
	Support	Shares common goals
		Cares for one another
		Helps each other at all times
		Keeps each other accountable
	Teamwork	Works together to achieve a goal
		Attachment
		Cohesive Bond
		Everyone is united
		Everyone knows their role
		Everyone is involved
		Cooperation
	Trust	Reliable relationship
		Honesty in C-A-P
		Safe C-A-P interaction
		C-A-P believes in each other
	Respect	Value each other
		In C-A-P everyone is appreciated
		In C-A-P everyone feels important
**Motivation**	Hard work	Effort
		Enthusiastic
		Achievements
	Passion	Feels attractive
		Feels driven
		Love what they do
		Inspires each other to be the best they can
	Enjoyment	Makes the activity fun
		Love the chosen activity
		Lives through the activity
**Over-involvement**	Demotivating	Oversteps boundaries
		One member expects more
		Poor behavior
	Demanding	Going above the limits
		Annoying
		Overly concerned
		Destructive behavior
	Controlling	Counting the achievements
		Focus on winning only
		Committed more than the other

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
