# Peer review of "The Initial Questionnaire Development in Measuring of Coach-Athlete–Parent Interpersonal Relationships: Results of Two Qualitative Investigations"

_ijerph, 2019, doi:10.3390/ijerph16132283_

Round 1

Reviewer 1 Report

Line 23: I don’t think the numbers are needed after the keywords

Line 40: there is a period of coaches.  Not sure if that is the end of the sentence or if the [4] reference is there on purpose.  Some is off with the punctuation

Line 152: two authors independently (no comma)

Line 153: Seems like additional language should be in here

For example: First article titles and their abstracts were reviewed.  Potential articles were then read next to determine inclusion in the study

Line 169: and were recruited

Author Response

Dear Reviewer, 

We have attached the revised manuscript version and the corrections that have been made according to your suggestions. 

Thank you.

Reviewer 2 Report

The article details the process carried out to develop a questionnaire that measures the coach-athlete-13 parent (C-A-P) interpersonal relationships in youth sports. For this, two qualitative studies were carried out with different participants, resulting in three main topics (group processes, motivation, and over-involvement) and 48 possible items.

Broad comments:

Although the study is extremely detailed and provides sufficient information about the process of developing the questionnaire, a quantitative validation of the questionnaire is recommended before publication of the article to verify its authentic validity. It would be necessary to apply the questionnaire to a sufficient sample and establish internal reliability using Cronbach's Alpha and perform a factorial analysis.

Author Response

(The authors gave the same response as above.)

Reviewer 3 Report

Dear Authors, 

The paper titled '' The Initial Questionnaire Development in Measuring 3 of Coach-Athlete-Parent Interpersonal Relationships: 4 Results of Two Qualitative Investigations'' is really interesting for me. 

Please, find some minor comments. 

- Since there is more than one researcher who analyzes the data it was better to test validity and consistency. 

In line 275 '' mean age 45 years participated in the study'' is this the age of all 10 coaches, 10 athletes, and 10 youth sports parents? why you didn't present the age of each group participants?

In line 276 can you explain what is the difference between ''coaches were highly educated and had a higher education degree''.

Based on the limitations mentioned in the study. There are no more comments from my side.

Author Response

(The authors gave the same response as above.)

Round 2

Reviewer 2 Report

The changes made are in line with the reviewer's request